# Enhanced B Cell Receptor Signaling Partially Compensates for Impaired Toll-like Receptor 4 Responses in LPS-Stimulated IκBNS-Deficient B Cells

**DOI:** 10.3390/cells12091229

**Published:** 2023-04-24

**Authors:** Monika Adori, Sharesta Khoenkhoen, Jingdian Zhang, Xaquin Castro Dopico, Gunilla B. Karlsson Hedestam

**Affiliations:** 1Department of Microbiology, Tumor and Cell Biology, Karolinska Institutet, 171 77 Stockholm, Sweden; 2Department of Medical Biochemistry and Biophysics, Division of Molecular Metabolism and Karolinska Institutet, 171 77 Stockholm, Sweden

**Keywords:** nfkbid, NF-κB, IκBNS, B cell activation, B cell receptor, TLR4, LPS, Lipid A, proliferation, cell cycling

## Abstract

Lipopolysaccharide (LPS) stimulates dual receptor signaling by bridging the B cell receptor and Toll-like receptor 4 (BCR/TLR4). B cells from IκBNS-deficient *bumble* mice treated with LPS display reduced proliferative capacity and impaired plasma cell differentiation. To improve our understanding of the regulatory role of IκBNS in B cell activation and differentiation, we investigated the BCR and TLR4 signaling pathways separately by using dimeric anti-IgM Fab (F(ab’)_2_) or lipid A, respectively. IκBNS-deficient B cells exhibited reduced survival and defective proliferative capacity in response to lipid A compared to B cells from wildtype (wt) control mice. In contrast, anti-IgM stimulation of *bumble* B cells resulted in enhanced viability and increased differentiation into CD138^+^ cells compared to control B cells. Anti-IgM-stimulated IκBNS-deficient B cells also showed enhanced cycle progression with increased levels of c-Myc and cyclin D2, and augmented levels of pCD79a, pSyk, and pERK compared to control B cells. These results suggest that IκBNS acts as a negative regulator of BCR signaling and a positive regulator of TLR4 signaling in mouse B cells.

## 1. Introduction

Polyclonal B cell stimulation leads to proliferation and differentiation of activated B cells into antibody-secreting plasma cells (PCs). Depending on the type of antigen, the response can be T cell-independent (TI) or T cell-dependent (TD). TD responses to protein-based antigens require cooperation of CD4^+^ T cells, while TI antigens can stimulate antibody production in the absence of cognate T cell help. TI antigens are classified as TI-1 and TI-2 types. TI-2 antigens, typically polysaccharides and polymers, contain highly repetitive epitopes that induce a strong immune response by the cross-linking of multiple B cell receptors (BCR). TI-1 antigens, such as lipopolysaccharide (LPS), are polyclonal B cell activators that can stimulate B cells directly through co-engagement of Toll-like receptors (TLRs) [1,2].

LPS, a widely used mouse B cell mitogen, is the major outer membrane component of Gram-negative bacteria, such as *Escherichia coli*, *Salmonella enterica*, and *Haemophilus influenzae* [3]. It has an evolutionarily conserved structure composed of three covalently linked, conformationally distinct parts: a hydrophobic lipid A backbone, a hydrophilic core oligosaccharide, and distal polysaccharide chains (O-antigen) [4]. The lipid A moiety is responsible for the endotoxic activity and immunostimulatory effects of LPS [5].

The innate immune system of mammals recognizes the lipid A motif through two receptor complexes, TLR4/MD-2 [6,7,8] and CD180 (RP105)/MD-1 [9,10], while the repetitive polysaccharide region of LPS cross-links BCRs. Thus, the optimal immune responses to LPS requires a concomitant engagement of both BCR and TLR4 [11]. Stimulation of these two receptors initiates signal transduction cascades that induce the translocation of nuclear factor kappa B (NF-κB) factors from their sequestered state in the cytoplasm to the nucleus, regulating the transcription of target genes [12,13,14].

In resting cells, homo-, or heterodimers of NF-κB transcription factors (NF-κB1 (p50), NF-κB2 (p52), RelA (p65), RelB, and c-Rel) are sequestered in the cytoplasm in an inactive form by the family of inhibitor κB (IκB) proteins [15]. Upon antigenic stimulation, IκB (classical pathway), or the p52 precursor p100 (alternative pathway) proteins are phosphorylated and subsequently degraded by the ubiquitin-proteasome system, releasing NF-κB dimers to translocate to the nucleus and regulate transcription [12,16]. Besides the classical cytoplasmic IκB proteins (IκBα, IκBβ, and IκBε; precursors p100/IκBδ and p105/IκBγ) that associate with, and thus inhibit, p50/p65 heterodimers, the nuclear atypical IκB proteins (BCL-3, IκBζ, IκBη, and IκBNS) may either enhance or suppress the transcriptional activity of target genes, depending on the cell type and stimulus [17]).

We previously described a mouse strain that lacks the atypical IκB protein, IκBNS, due to a non-sense mutation in the *nfkbid* gene (*bumble* mice) [18]. IκBNS was reported to bind to p50 homodimers, p52, and the Rel proteins [17,19]. *Bumble* (henceforward referred to as *bmb*) mice exhibit reduced frequencies of splenic marginal zone (MZ) B cells and peritoneal B-1 cells, and show an impaired humoral immune response to TI antigens [18,20], which are related to metabolically dysregulated and diminished PC differentiation [21,22,23].

In the present study, we used *bmb* mice to investigate the regulatory role of IκBNS in TI B cell activation in vitro. Specifically, we investigated BCR and TLR4 stimulation separately using dimeric anti-IgM Fab (F(ab’)_2_) (hereinafter anti-IgM) or the lipid A component of LPS, respectively. We demonstrate that *bmb* B cells displayed impaired survival and altered proliferative capacity upon lipid A stimulation compared to B cells from wt mice. In contrast, in the absence of IκBNS, anti-IgM stimulation resulted in enhanced B cell survival, which was associated with increased blast formation and a higher proportion of B cells that differentiated into CD138^+^ plasma cells. Furthermore, in response to anti-IgM stimulation, *bmb* B cells displayed accelerated proliferation and cell cycle progression with a partial G2 arrest, which correlated with elevated c-Myc and cyclin D2 levels and enhanced phosphorylation of early downstream components of BCR signaling, such as CD79, ERK, and Syk. These results offer an enhanced understanding of the role of IκBNS in LPS-induced B cell activation.

## 2. Materials and Methods

### 2.1. Mice 

IκBNS-deficient *bumble* mice, generated by ENU mutagenesis of C57BL/6J mice, and B cell specific IκBNS conditional knock-out mice have been described previously [18,24]. Mice were bred and maintained at Karolinska Institutet animal research facility KM-A under ethical permit number 20513–2020 (ethical statement approval date: 11 February 2021) (Jordbruksverket). Studies were conducted with the approval of the Committee for Animal Ethics (Stockholms Norra Djurförsöksetiska nämnd), and all experimental procedures were performed in accordance with institutionally approved protocols. Genotyping of the IκBNS conditional knock-out offspring was carried out by PCR analysis of genomic DNA, as described previously [24]. 8–14-week-old, age- and sex-matched mice were used for the experiments.

### 2.2. B Cell Isolation and In Vitro Cultures

Splenic single-cell suspensions were prepared using 70 μm cell strainers and syringe plungers. Red blood cells (RBC) were lysed using isotonic lysis solution (1.5 M ammonium chloride, 100 mM sodium bicarbonate, and 10 mM EDTA, pH 7.2–7.4). Cell suspensions were washed twice in Ca^2+^- and Mg^2+^-free PBS (Sigma-Aldrich, St. Louis, MO, USA). B cells were isolated by negative selection using the EasySep Mouse B Cell Isolation Kit (STEMCELL Technologies, Vancouver, BC, Canada) according to the manufacturer’s protocol. B cell purity was ≥95% based on CD19 staining and flow cytometry. Cells were resuspended in RPMI 1640 medium containing 2.05 mM L-glutamine (HyClone, Logan, UT, USA), supplemented with 10% fetal bovine serum (FBS) (HyClone, Logan, UT, USA), penicillin (100 IU)-streptomycin (100 μg/mL) (Sigma-Aldrich, St. Louis, MO, USA), and β-mercaptoethanol (0.05 mM) (Gibco, Waltham, MA, USA) (complete RPMI medium), unless otherwise stated.

Purified B cells (at 2.5 × 10^5^ or 5 × 10^5^ cells per well, as indicated) were cultured in 48- or 96-well flat-bottom plates (Corning, NY, USA) in RPMI medium, and were stimulated with goat anti-mouse IgM F(ab’)_2_ (#115-006-020; Jackson ImmunoResearch Laboratories, West Grove, PA, USA), LPS (from *Escherichia coli* serotype 0111:B4; Sigma-Aldrich, St. Louis, MO, USA), or monophosphoryl lipid A from *Escherichia coli* serotype R515 (Re) (Enzo Life Sciences, Farmingdale, NY, USA) in the indicated concentrations for the indicated times, or left unstimulated.

### 2.3. CTV Labeling 

To monitor cell division, purified B cells were labeled with 2.5 μM CellTrace Violet (CTV) dye (Thermo Fisher Scientific, Waltham, MA, USA) in PBS at 37 °C for 20 min according to the manufacturer’s protocol prior to stimulation. Cell divisions were analyzed with the proliferation platform of FlowJo v10.7.2 software (Tree Star, Ashland, OR, USA).

### 2.4. Cell Cycle Analysis 

For cell cycle analysis at the end of the culture period, the stimulated B cells were harvested and counted. Cell numbers and viability were determined with trypan blue exclusion using a Countess II automated cell counter (Thermo Fisher Scientific, Waltham, MA, USA). After counting, cells were washed, then resuspended in 0.5 mL of ice-cold PBS. To the cell suspension, 1 mL of ice-cold absolute ethanol was added dropwise while the tube was gently vortexed. Cells were kept on ice for at least 30 min, then stored at −20 °C until further procedures. For propidium iodide (PI) staining, ethanol-fixed cells were washed twice with ice-cold PBS and then, based on the cell numbers, were resuspended and incubated in PI staining solution (500 μL/1 M cells; 20 μg/mL propidium iodide (#P4170, Sigma-Aldrich, St. Louis, MO, USA), 50 μg/mL RNAse A (#R6513, Sigma-Aldrich, St. Louis, MO, USA), and 0.1% Triton X-100 in PBS) for 30 min at 37 °C. Alternatively, stimulated B cells were pulsed with 10 μM thymidine analog 5-Bromo-2′-deoxyuridine (BrdU) (#B5002, Sigma-Aldrich, St. Louis, MO, USA) for the last 1 h of culture. After harvesting, counting, and fixing (as described above), DNA were denatured with 2 M HCl/0.5% Triton X-100 for 30 min at room temperature (RT) following neutralization with 0.1 M Sodium Tetraborate (Na_2_B_4_O_7_), pH 8.5 for 15 min at RT. Cells were then stained with FITC-conjugated anti-BrdU antibody (#556028, BD Biosciences, Franklin Lakes, NJ, USA) in PBS/0.5% BSA/0.5% Tween 20 for 30 min at RT. After washing in PBS/0.5% BSA, stained cells were incubated with PI solution for 1 h at 37 °C [25,26]. To distinguish cells in G0 stage, intracellular staining of Ki-67 (Alexa Fluor 488-conjugated anti-mouse Ki-67 (#16A8, BioLegend, San Diego, CA, USA)) was performed using the BD transcription factor buffer set (BD Biosciences, Franklin Lakes, NJ, USA), according to the manufacturer’s protocol. To discriminate mitotic cells in the G2/M phase, Alexa Fluor 488-conjugated anti-pHH3 (Ser10) (#D2C8, Cell Signaling Technology, Beverly, MA, USA) was used, together with PI [27]. Cell acquisition was performed on a BD FACSCelesta instrument (BD Biosciences, Franklin Lakes, NJ, USA).

### 2.5. Phosphoflow Cytometry 

For detection of phosphorylated intracellular proteins, phosphoflow cytometry was performed, as described previously [28]. Briefly, 5 × 10^5^ B cells were plated in RPMI medium supplemented with 2% heat-inactivated FBS at 37 °C in 96-well flat-bottom TC plates (Corning, NY, USA). Cells were stimulated with 25 μg/mL goat anti-mouse IgM F(ab’)_2_ fragments (Jackson ImmunoResearch Laboratories, West Grove, PA, USA) for 5 min for pCD79a, pSyk, and pERK, or were left unstimulated. Cells were fixed and permeabilized for 10 min at 37 °C with the FoxP3/Transcription Factor staining Fixation/Permeabilization kit (eBioscience, San Diego, CA, USA). After two washes with Perm/Wash solution, cells were labeled with surface B cell marker (B220 BV421 or BV786; BD Biosciences, Franklin Lakes, NJ, USA) for 20 min on ice. Intracellular staining with the phospho-target Abs was performed in Perm/Wash buffer for 30 min at RT. The following phosphoprotein-specific antibodies were used: pCD79a (Tyr182) (clone D1B9) Alexa Fluor 647 (Cell Signaling Technology, Beverly, MA, USA), pSyk (Tyr348) (clone I120–722) PE, pERK 1/2 (Thr202/Tyr204) (clone 20A) PE (both from BD Biosciences). After staining and washing, samples were measured on a FACSCelesta instrument (BD Biosciences) and analyzed using FlowJo v10 (Tree Star, Ashland, OR, USA).

### 2.6. Flow Cytometry

Dead cells were discriminated using a Live/Dead Fixable Far Red Dead Cell Stain Kit (Invitrogen, Carlsbad, CA, USA), following the manufacturer’s protocol. To block nonspecific antibody binding to Fc receptors, cells were incubated with anti-CD16/32 antibody (2.4G2; BD) in PBS-2% FBS (FACS buffer), and they were then stained with fluorochrome-conjugated monoclonal antibodies at 4 °C for 30 min. Cell populations are pre-gated as singlet (FSC-W vs. FSC-A) and lymphocyte (SSC-A vs. FSC-A) subsets prior to further gating, as indicated in figure legends. Samples were acquired on a BD FACSCelesta or an LSRII flow cytometer, and data were analyzed with FlowJo software (Tree Star). The following fluorochrome-labeled antibodies were used: B220 (clone RA3-6B2) BV786 and CD138 (clone 281-2) BV650 from BD, IgM (clone RMM-1) Alexa Fluor 647, TLR4/MD-2 (clone MTS510) PE, CD180 (RP105) (clone RP/14) PE, MD-1 (clone MD-113) PE (all from BioLegend, San Diego, CA, USA), and CD86 (clone GL1) PE (eBioscience, San Diego, CA, USA). For follicular B cell (FoB) sorts, cells were labeled with Live/Dead Fixable Aqua Dye (Invitrogene, Carlsbad, CA, USA) following surface staining with the mixture of B220 (clone RA3-6B2) BV786, CD93 (clone AA4.1) BV421, CD21/35 (clone 7G6) FITC (all from BD), and CD23 (clone B3B4) Alexa Fluor 647 (BioLegend, San Diego, CA, USA) antibodies. Cells were sorted on a BD FACSAria Fusion cell sorter (BD).

### 2.7. Seahorse Mitochondrial Stress Test 

To measure mitochondrial respiration of FoB cells in real time, a Seahorse XFe96 Extracellular Flux Analyzer with a Seahorse XF Cell Mito Stress Test Kit (Agilent Technologies, Santa Clara, CA, USA) was used [22]. After 3 days of BCR stimulation, FoB cells were harvested, counted using trypan blue exclusion, and washed with Seahorse XF RPMI assay medium. After this, 2 × 10^5^ cells/well were seeded in 40 μL assay medium on a poly-D-lysine-coated (Sigma-Aldrich, St. Louis, MO, USA) XF 96-well cell culture microplate. The plate was centrifuged at 300× *g* for 1 sec without break, rotated, and centrifugation was repeated. After this, 140 μL of assay medium was added per well, and the plate was rested for 40 min in a 37 °C, non-CO_2_ incubator. During the assay, the following compounds (final concentrations) were injected sequentially into the wells: oligomycin (1.264 μM) (Sigma-Aldrich, St. Louis, MO, USA; #O4876), FCCP (2 μM) (Sigma-Aldrich, St. Louis, MO, USA; #C2920), and rotenone, together with antimycin A (both 0.5 μM; Sigma-Aldrich, St. Louis, MO, USA; #R8875 and #A8674, respectively). Oxygen consumption rate (OCR) was measured three times, every three minutes from each well, before and after each injection. OCR values were normalized to protein concentrations (determined by BCA assay (Thermo Fisher Scientific, Waltham, MA, USA)) and analyzed using the Wave Desktop software 2.6.1 (Agilent Technologies, Santa Clara, CA, USA). 

### 2.8. Western Blot Analysis

For whole cell lysates, cell pellets were lysed in RIPA buffer supplemented with protease and phosphatase inhibitors (cOmplete and PhosSTOP, respectively; Roche, Basel, Switzerland). Proteins were subjected to NuPAGE 4–12% Bis-Tris gel (Invitrogen, Carlsbad, CA, USA) and transferred to polyvinylidene difluoride (PVDF) membranes (0.45 μm, Immobilon-P (Sigma-Aldrich, St. Louis, MO, USA)). Non-specific binding was blocked with PBS containing 5% non-fat milk and 0.1% Tween 20 for 1 h at RT, followed by overnight incubation at 4 °C with the primary antibodies diluted in blocking buffer. After washing, membranes were incubated for 1 h with the appropriate horseradish peroxidase (HRP)-conjugated secondary antibodies. Protein detection was performed using ECL Plus Western Blotting Substrate reagent (Thermo Fisher Scientific, Waltham, MA, USA) and visualized by ChemiDoc XRS+ Imaging System (Bio-Rad, Hercules, CA, USA). The following antibodies were used: c-Myc (clone D84C12), Cyclin D2 (clone D52F9), β-actin mouse mAb (clone 8H10D10), HRP-conjugated polyclonal goat anti-rabbit IgG (#7074) (all from Cell Signaling Technology, Beverly, MA, USA), HRP-conjugated polyclonal goat anti-mouse IgG (#1030-05; Southern Biotech, Birmingham, AL, USA).

### 2.9. Statistics

Differences between groups were analyzed by unpaired *t* test or Mann–Whitney *t*-test using GraphPad Prism v8. Statistical significance is indicated with ns (non-significant) for *p* > 0.05, * for *p ≤* 0.05, ** for *p ≤* 0.01, *** for *p ≤* 0.001, and **** for *p ≤* 0.0001. 

## 3. Results

### 3.1. IκBNS-Deficient B Cells Display Impaired Survival, Proliferation, and Differentiation in Response to TLR4 Agonists

LPS induces proliferation and differentiation of mouse B cells into antibody-secreting plasma cells (PCs) [29] and exerts pro-survival activity on both immature and mature B cells [30,31]. LPS stimulates dual BCR and TLR4 signaling by engaging the BCR through its polysaccharide and TLR4 through its lipid A component [11]. The nuclear IκBNS regulator is downstream of both the BCR and TLR4 pathways, modulating the NF-κB response (Figure 1A). In previous studies, we showed that B cells from IκBNS-deficient *bmb* mice display reduced capacity to proliferate and differentiate in response to LPS in vitro and in vivo [22,23]. However, the respective roles of the two pathways were not investigated. 

Here, to improve our understanding of the role of IκBNS in LPS-stimulated B cells, we used lipid A to stimulate TLR4 in the absence of BCR engagement and examined survival, proliferation, and the capacity of the B cells to differentiate into PCs in vitro. Following three days of culture in the presence of either LPS or lipid A, we found significantly fewer viable cells in the IκBNS-deficient B cell cultures compared to control cultures, and the decline was more pronounced following lipid A stimulation (Figure 1B). 

Following mitogenic stimulation, B cells enlarge in size, which can be monitored by flow cytometry as an increased forward (FSC) vs. side scattered (SSC) profile (Appendix A). Following three days stimulation with either LPS or lipid A, we observed significantly reduced frequencies of live blasted cells in *bmb* B cell cultures compared to controls (85% blasted cells in wt vs. 70–80% in *bmb*). The defect was more striking in the case of lipid A stimulation, because only 15–25% of the viable *bmb* cells showed blasted phenotype in contrast to the 60–70% of wt cells (Figure 1C).

Flow cytometric analysis of viability using a fixable live/dead dye showed a modest, but significant, reduction in the frequencies of live cells (viability marker^low^) (10–15% reduction) in LPS-stimulated *bmb* cultures, compared to in wt cultures, with a greater decline (40–50%) after lipid A stimulation (Figure 1D).

Proliferation of LPS- or lipid A-stimulated B cells was traced by the dilution of the fluorescent division tracking dye, CellTrace Violet (CTV). As reported previously [23], the absence of IκBNS resulted in an altered proliferative capacity to LPS, with higher frequencies of cells in the non-divided fraction and the early-mid divisions (divisions 3–5), with only a small proportion of cells reaching the last division (division 7). The frequency of viable cells was markedly reduced in lipid A-stimulated *bmb* cell cultures, compared to in control cultures. The frequencies of cells in the non-divided fraction and in the early-mid divisions (divisions 1–4) were significantly higher in *bmb* cultures, with only a small portion reaching the later divisions (beyond 5) compared to control cultures (Figure 1D). The reduced viability and proliferative response to lipid A was irrespective of the lipid A concentration used for stimulation (Appendix A).

To compare the effect of IκBNS for LPS- or lipid A-stimulated PC differentiation, we analyzed cells by flow cytometry after three days in vitro stimulation. Dual staining of B220 and the plasma blast (PB)/PC marker, CD138 (syndecan-1), showed that both stimuli promoted PC differentiation of wt B cells, while *bmb* B cells failed to generate PCs in response to either TLR4 ligand. In agreement with our previous findings [22], the frequency of PBs was greater in LPS-stimulated *bmb* cultures than in control cultures, while after lipid A treatment, there was no significant difference in the frequency of PBs between the wt and *bmb* B cell cultures (Figure 1E).

B cells recognize LPS and lipid A by two receptor complexes, the TLR4/myeloid differentiation factor-2 (MD-2) and CD180 (RP105)/MD-1 [7]. To determine whether the reduced responsiveness of *bmb* B cells to LPS and lipid A resulted from differences in cell surface levels of these receptors, we analyzed their expression by flow cytometry on freshly isolated B cells. The expression of both the TLR4/MD-2 complex and CD180 were comparable in wt and *bmb* B cells, while MD-1 level was slightly elevated in *bmb* B cells compared to in control cells (Figure 1F). This experiment showed that the reduced responsiveness of *bmb* B cells to TLR4 stimulation in terms of viability, blasting, proliferation, and PC differentiation was not the result of reduced receptor expression.

### 3.2. IκBNS-Deficient B Cells Display Improved Survival, Proliferation, and Differentiation in Response to BCR Stimulation

*Bmb* follicular B (FoB) cells display elevated surface IgM BCR expression (Appendix A and [18,32]). However, the potential effect of this on B cell activation and proliferation was previously not studied in detail. To explore this, we stimulated wt and *bmb* B cells with dimeric anti-IgM Fab (F(ab’)_2_) (hereinafter anti-IgM) [33] and determined cell viability and proliferation by flow cytometry. Following a three-day stimulation protocol, we observed increased cell numbers and viability in *bmb* B cell cultures compared to in control cultures **(**Figure 2A,B). The enhanced survival in the absence of IκBNS was associated with the augmented frequency, size, and granularity of live blasted cells, as shown by the FSC/SSC profiles (Figure 2C). Notably, a large proportion (40–50%) of the blasted *bmb* cells was CD138^+^, while only 15–25% of wt cells expressed CD138 (Figure 2D). The proliferative capacity of the IκBNS-deficient B cells was also altered upon BCR stimulation, with a greater proportion of *bmb* B cells reaching division 2, but with very few cells reaching the last division, compared to control B cells (Figure 2E).

To assess whether the strength of BCR engagement influenced IκBNS-deficient B cell proliferation, we stimulated the B cells with different doses of anti-IgM ranging from 1 to 50 μg/mL for three days. We found that an increasing anti-IgM dose resulted in enhanced B cell expansion, irrespective of IκBNS, with higher counted live cell numbers in *bmb* compared to in wt cultures up until 10 μg/mL anti-IgM. Notably, IκBNS-deficient B cells exhibited enhanced spontaneous cell death, as the frequency of live cells was significantly lower in the non-stimulated *bmb* cell cultures, compared to in wt cells (Appendix A). Similar to LPS or lipid A stimulation, IκBNS-defective B cells activated with 5 μg/mL or higher doses of anti-IgM showed altered proliferation, as the frequencies of cells were significantly lower in the non-divided fraction and the earlier divisions (division 1, or division 2 in the case of the highest anti-IgM concentration). At 5–50 μg/mL anti-IgM concentrations, we observed remarkably more *bmb* cells in division 3, while the cell frequencies reduced beyond the 4th divisions and were lower than in wt cultures (Appendix A). We next examined whether the duration of BCR ligation enhanced the proliferation rate of IκBNS-deficient B cells. After 5 days of anti-IgM stimulation, despite increased viability, *bmb* B cells displayed defective proliferation with a significantly smaller proportion of cells progressing beyond division 4, while wt cells reached 7 divisions (Appendix A). 

B cell activation is associated with the upregulation of co-stimulation molecules. We found that 24 h of anti-IgM stimulation increased CD86 surface expression in both *bmb* and wt B cells, although the levels were significantly lower in *bmb* cells. Stimulation for 48 h resulted in a further increase of surface CD86 in wt cells, while this was not observed in *bmb* cells. There was no further upregulation of CD86 after 72 h of stimulation in either of the cultures (Figure 2F). Collectively, these results demonstrate that B cells lacking IκBNS displayed enhanced early responses to anti-IgM in terms of viability and blast formation, while activation and proliferation were perturbed.

### 3.3. Accelerated Cell Cycle Progression of BCR-Stimulated IκBNS-Deficient B Cells Is Associated with a Partial Block in G2

The observed perturbation in *bmb* B cell proliferation upon BCR ligation prompted us to investigate cell cycle progression following anti-IgM stimulation. First, we examined the expression of the proliferation-associated nuclear protein, Ki-67, which is present during all active phases of the cell cycle (G1, S, G2, and M), but is absent from quiescent (G0) cells [34]. Following 48 and 72 h BCR stimulation, we found elevated frequencies of Ki-67^+^ cells in *bmb* cultures compared to in wt (Figure 3A). To discriminate the different cell cycle phases (Sub-G0/G1, G0/G1, S, G2/M), we applied bromodeoxyuridine (BrdU) to mark the cells in S phase in combination with propidium iodide (PI) staining, as an indicator of DNA content, and analyzed B cell cultures from *bmb* and wt mice at 48 and 72 h following anti-IgM stimulation. Cells with lower DNA content (< 2N; Sub-G0/G1) were defined as apoptotic cells. The percentage of the sub-G0/G1 population in *bmb* cultures was significantly reduced at both time points, in comparison with the percentage in wt cultures (Appendix A). After excluding the sub-G0/G1 population, we evaluated the proportions of each cell cycle stage in the live populations.

IκBNS-deficient *bmb* B cells entered the cell cycle at an elevated rate compared to control cells, as demonstrated by a significant decrease in cell frequencies in the G0/G1 phase (BrdU negative cells with a 2N DNA content), associated with increased frequencies in both BrdU positive cells, marking S phase, and BrdU negative cells with 4N DNA content, as G2/M phase after 72 h anti-IgM stimulation. At the 48-h time point, we found lower frequencies of *bmb* cells in the G2/M stage, compared to wt control cells (Figure 3B). To further distinguish the G2 and M phases, we stained cells for both DNA content (PI) and histone H3 phosphorylated at serine 10 (pHH3), which marks cells in mitosis. At 72 h post-stimuli, we found a reduced proportion of pHH3 positive, mitotic (M) cells within the G2/M population in *bmb* cell culture compared to the proportion in wt cells. Therefore, *bmb* B cells appeared to accumulate in the G2 phase, with fewer cells entering the mitotic stage (Figure 3C). 

Cell cycle progression is a tightly regulated process, controlled by multiple proteins (cyclins, cyclin-dependent kinases). The NF-κB target transcription factor, c-Myc, plays a crucial role in BCR-triggered cell proliferation by controlling cell cycle entry into the G1 phase, or the transition from the G1 to the S phase via cyclin D2 induction (rev. in [35]). Time course analysis of c-Myc and cyclin D2 expression levels by Western blot showed significantly higher levels of c-Myc protein in *bmb* B cells compared to in wt cells, after both 48 and 72 h BCR stimulation, with a more prominent difference at 72 h. Cyclin D2 levels were comparable in both strains at 48 h, while after 72 h of anti-IgM treatment cyclin D2 was slightly downregulated in wt but remained high in *bmb* B cells (Figure 3D). Taken together, these results show that anti-IgM-stimulated *bmb* B cells exhibited an accelerated cell cycle entry, which was associated with elevated c-Myc and cyclin D2 expression compared to control B cells. However, the proliferating *bmb* cells were unable to complete the cycling program, because a higher proportion of cells were found in G2, and lower proportion of cells were found in the M stage, compared to similarly stimulated wt B cells.

### 3.4. Increased Mitochondrial Respiration in BCR-Stimulated IκBNS-Deficient B Cells

Cell growth and proliferation is an energy-consuming process, which requires increased mitochondrial ATP production to support the energetic demand of the cells [36,37,38,39,40]. In a recent study, we showed that the absence of IκBNS in LPS-stimulated B cells resulted in elevated oxidative phosphorylation (OXPHOS) [22]. To investigate mitochondrial function in BCR-stimulated, proliferating *bmb* B cells, we measured the mitochondrial oxygen consumption rates (OCR) by Seahorse analyzer. Because *bmb* mice have similar FoB cell numbers as wt mice but reduced marginal zone (MZ) B cell frequencies [18], we used flow cytometry-sorted FoB cells to investigate mitochondrial respiration. Following 72 h of anti-IgM stimulation, *bmb* FoB cells showed significantly increased basal, maximal oxygen consumption rate (OCR), spare respiratory capacity, and elevated ATP-coupled respiration/OCR, compared to wt FoB cells (Figure 4). These results demonstrate that the loss of IκBNS results in enhanced mitochondrial activity in anti-IgM-stimulated B cells.

We next sought to explore whether the enhanced survival and accelerated cell cycle in the absence of IκBNS was the result of increased BCR signaling. We performed intracellular FACS analysis for early phosphorylation events triggered by BCR cross-linking, such as pCD79a, pSyk, and pERK. We found higher pCD79 levels in unstimulated *bmb* compared to control cells, while the basal levels of pSyk and pERK in unstimulated wt and *bmb* B cells were similar. Following anti-IgM treatment, the levels of all examined phosphoproteins were elevated compared to unstimulated cells, and they were significantly higher in *bmb* B cells compared to in wt cells (Figure 5). These results suggest that IκBNS regulates early molecular events downstream of BCR signaling.

### 3.5. Dual Stimulation through BCR and TLR4 Has Synergistic Effect on Viability and Proliferating Capacity in IκBNS-Deficient B Cells

It was shown previously that simultaneous stimulation through BCR and TLR enhances B cell activation and proliferation additively [41,42]; however, a persistent signal via BCR can abolish the LPS-induced PC differentiation [43]. Comparing the effects of the single and combined stimulation on B cell viability and proliferation, we found that simultaneous engagement of BCR and TLR4 led to increased cell survival in both wt and *bmb* cultures, with an elevated synergistic effect in *bmb* B cell cultures. IκBNS-deficient B cells also showed increased proliferative ability (Figure 6A,B). Monitoring the expression of activation marker CD86 showed a synergic effect of the dual stimulation in both strains, as the frequency of CD86^+^ cell and MFI values for CD86 were significantly increased after anti-IgM plus lipid A treatment (Figure 6C, Appendix A). Thus, combined stimulation with lipid A fragment and anti-IgM led to increased viability and proliferating capacity of *bmb* B cells and elevated CD86 expression.

To examine if increased responses to anti-IgM stimulation were a consequence of enhanced surface IgM expression in *bmb* FoBs [18,32], we used conditional knock-out mice deleted for *nfkbid* exclusively in the B cell lineage (referred to as *nfkbid*^B−^). B cells in these mice express comparable levels of surface IgM as control *nfkbid*^B+^ B cells [24]. Following 72 h in vitro anti-IgM stimulation, in *nfkbid*^B−^ B cell cultures we observed significantly higher live cell numbers compared to in control cultures (Figure 7A). Consistent with this, discrimination of viable cells by flow cytometry showed an elevated frequency of live cells (viability dye^low^) in the conditional knock-out cell culture (Figure 7B). In contrast, TLR4 stimulation with LPS or lipid A resulted in significantly reduced B cell survival in the *nfkbid*^B−^ cultures compared to in the controls, which was more remarkable in the case of lipid A (Figure 7A,B). Furthermore, in response to TLR4 activation, conditional knock-out B cells failed to differentiate into PCs, as B220 CD138 dual staining showed, while there was no reduction in the B220^+^ CD138^+^ PB population, a rather modest increase in the case of LPS stimulation (Figure 7C). Upon anti-IgM stimulation, increased frequencies of CD138^+^ cells were generated in the *nfkbid*^B−^ cultures compared to in the *nfkbid*^B+^ controls (Figure 7D). Similar to IκBNS-deficient *bmb* B cells, conditional knock-out B cells displayed altered proliferative ability in response to all three stimuli (Figure 7E). These results indicate that the elevated response of *bmb* B cells to anti-IgM is not due to the increased surface IgM expression. Therefore, we conclude that the reduced responsiveness to TLR4 stimulation, as well as the enhanced response to anti-IgM stimulation, are B cell intrinsic features.

## 4. Discussion

The innate immune system has evolved to serve as the first line of defense against pathogens. This process involves the recognition of pathogen-associated molecular patterns (PAMPs) on pathogens by pattern recognition receptors (PRRs) expressed by host cells. LPS is a well-characterized PAMP in the outer membrane of Gram-negative bacteria, which induces B cell activation by the dual engagement of TLR4 and BCR, via the small lipid A moiety and the connected polysaccharide chain, respectively. Synergy of the two signaling pathways leads to the activation of NF-κB cascade, engendering a protective response against capsular bacteria [11].

In the present study, we investigated the role of IκBNS in BCR- and TLR4-induced B cell activation using the *bmb* mouse strain deficient for the NF-κB pathway regulator, IκBNS protein [18]. We applied monophosphoryl lipid A and bivalent anti-IgM F(ab’)_2_ fragment (anti-IgM) to study the two pathways separately.

We report that in the absence of functional IκBNS, B cells show reduced survival after lipid A or LPS stimulation in vitro. Viable cells exhibited altered cell proliferation, with increased cell frequencies in the early, and reduced frequencies in the late divisions. Similar to the LPS response [23], lipid A treatment did not promote PC differentiation of *bmb* B cells.

In contrast, anti-IgM stimulation induces elevated *bmb* cell survival, cell growth, and blast formation associated with a higher frequency of CD138^+^ B cells in comparison with wt cells.

Moreover, following BCR stimulation, *bmb* B cells showed a perturbed proliferation and accelerated cell cycle entry, accompanied by a partial block in the G2 phase. The enhanced proliferation is associated with elevated levels of c-Myc and cyclin D2.

Furthermore, anti-IgM-stimulated *bmb* B cells displayed increased phosphorylation of early BCR signaling molecules, such as CD79a, Syk, and ERK. Studies on IκBNS B cell conditional knock-out mice revealed that the enhanced BCR response is B cell intrinsic and not the consequence of elevated surface IgM expression that was found in *bmb* B cells.

Studies using knock-out mice have demonstrated the critical role of NF-κB transcription factors in B cell proliferation. Mice with a single deficiency in NF-κB1 (p50) [44,45], c-Rel [46], or Rel-B [47], and the double mutant *nfkb1*^−/−^*c-rel*^−/−^ mice [48,49] showed severely impaired proliferative defects in response to mitogenic stimuli.

Here, in agreement with our previous study [23], we showed that LPS- and lipid A-stimulated *bmb* B cells exhibited an altered proliferation profile, with an initial acceleration followed by a decline, by which less cells achieve the later divisions. 

Although TLR4 is the commonly referred LPS receptor, LPS binding requires a non-covalent association between TLR4 and myeloid differentiation factor 2 (MD-2) [7,50,51]. MD-2 binds the lipid A part of LPS, inducing the homodimerization of the TLR4/MD-2 complex and subsequent activation of the downstream signaling pathways of TLR4 [13,52,53].

TLR4 belongs to the family of PRRs characterized by two major domains: the extracellular leucine-rich repeat motif (LRR) and the intracellular Toll/interleukin 1 (IL-1) receptor (TIR) signaling domain [13].

Following ligand binding, TLR4 mediates myeloid differentiation primary response gene 88 (MyD88)-dependent and Toll/IL-1R (TIR) domain-containing adaptor-inducing interferon-β (TRIF)-dependent (or MyD88-independent) signaling pathways. The MyD88-mediated pathway progresses through the IL-1R-associated kinases (IRAKs), and TRAF6 cascade subsequently leads to the activation of the c-Jun N-terminal kinase (JNK)/p38 mitogen-activated protein (MAP) kinase family [54] and an early wave of NF-κBs. This pathway is critical for the induction of pro-inflammatory cytokines (IL-1β, TNFα, IL-6). The MyD88-independent pathway requires the internalization of TLR4, which activates the TRIF/interferon regulatory factors (IRF) 3 axis, leading the production of IFNβ and IP-10 as well as a delayed NF-κB activation [55].

Functional LPS receptor expression is essential for proper B cell activation and antibody production. Even though B cells express relatively low TLR4/MD-2 in comparison with macrophages, analysis of mutant mice demonstrate that this complex is indispensable for B-cell responses to LPS [7,56].

Besides TLR4, the TLR-like molecule CD180 (RP105) was reported to be implicated in LPS-triggered B cell activation [57]. Similar to TLR4, CD180 has a conserved extracellular leucine-rich repeat (LRR) domain; however, it lacks the intracellular Toll-IL-1R (TIR) signaling domain. The expression and function of CD180 are dependent on the co-expression of MD-2 homologue MD-1 protein [58,59]. The two complexes together are crucial for the appropriate LPS or lipid A-induced B cell responses.

In the current study, analysis of the expression levels of TLR4/MD-2 and CD180/MD-1 complexes on purified B cells from wt and *bmb* mice did not reveal reduced expression of the LPS receptors in *bmb*, which indicates that the proliferative defect in response to LPS or lipid A is not due to decreased receptor expression. However, normal receptor expression or proper physical interaction between receptor and ligand is not always manifested in an activation signal, as showed in lipid A-stimulated MyD88^−/−^ B cells, which failed to proliferate or up-regulate the activation marker CD86 [60]. Thus, to elucidate the survival and proliferation defect in LPS-/lipid A-stimulated *bmb* B cells, further evaluation of LPS/lipid A-triggered signaling pathways is needed.

Besides its potent mitogenic effect, LPS promotes the survival of immature and mature B cells [30,61] by preventing apoptosis through the NF-κB-dependent regulation of pro/anti-apoptotic molecules [62,63]. LPS- or lipid A-stimulated *bmb* B cells, however, showed reduced viability, while BCR stimulation by anti-IgM triggered an elevated survival of *bmb* B cells, indicating that IκBNS has an opposite regulatory role during the TLR4- and BCR-induced cell survival, namely a positive in TLR4- and a negative in BCR-stimulated.

Upon antigen activation, B cells differentiate into antibody secreting PCs, which can be identified by the high surface expression of CD138 (syndecan-1) [64,65] in conjunction with low levels of B220 [66]. As we previously reported, LPS-stimulated *bmb* B cells failed to generate PCs, both in vitro and in vivo, as the PC differentiation was suspended in the premature CD138^+^ B220^+^ pre-PB, PB stages [22,23]. This was also the case with the TLR4 agonist, lipid A.

Divalent anti-IgM F(ab’)_2_ fragment, which mimics antigen for BCR, promotes B cell activation, blast formation, and proliferation. However, proper PC differentiation and antibody synthesis requires the help of cognate, antigen-primed T cells [67,68].

Remarkably, we observed a perturbed proliferation pattern in the anti-IgM-stimulated *bmb* cultures, as in the LPS- or lipid A-treated cultures. Specifically, the elevated cell frequencies in mid divisions were accompanied by reduced cell frequencies in the later divisions. These results suggest a general B cell proliferation deficit in response to the examined TI stimuli, in the absence of IκBNS.

Furthermore, unexpectedly, after 3 days of anti-IgM activation, a remarkable proportion of the live *bmb* cells were found to express the PC marker CD138 (Figure 2D). It has been proposed that CD138 (syndecan-1) promotes the survival of mature PCs [69], as well as multiple myeloma cells, by binding of anti-apoptotic factors, growth factors, and pro-survival cytokines, such as APRIL, or IL-6 through its heparan sulfate moieties [70,71]. In accordance with this, IgM-stimulated *bmb* cell cultures contained elevated numbers and frequencies of viable cells. Notably, the elevated anti-IgM response in the conditional knock-out B cells, which shows comparable surface IgM expression levels to the control B cells, indicates that the enhanced BCR-induced response in *bmb* B cells is not a consequence of elevated surface IgM expression on these cells.

The B cell proliferation defect in the single p50 and c-Rel mutant mice is associated with a block in the G1 phase [72], while in the double knock-out it was manifested as a failure of cell growth. This was due to impaired c-Myc expression, which normally is induced by NF-κB1/c-Rel dimers via a PI3K-dependent pathway [73]. Besides its function in cell size regulation, whose process is uncoupled from cell division, c-Myc has a crucial role in cell cycle control [35,74,75,76].

Cell cycle is strictly regulated by the heterodimeric complexes of cycling-dependent kinases (CDKs), cyclins, and inhibitors of CDKs [77]. c-Myc plays a crucial role in the BCR-triggered cell proliferation by controlling cell cycle entry into the G1 phase [78,79,80], and the transition from the G1 to the S phase via the direct induction of cyclin D2 [81,82,83].

Although IκBNS is important in LPS- [23], or anti-IgM-induced B cell proliferation, its possible role in cell cycle progression has not been studied.

Notably, BCR-triggered *bmb* B cells exhibited increased cell size, which was associated with accelerated cell cycle entry and transition to the S phase, and a concurrent partial block in the G2 phase with reduced entry into mitosis. The latter is reflected by the decreased frequency of cells that reached later divisions. These features coincided with elevated c-Myc and cyclin D2 expression levels. 

Akkaya et al. reported that BCR-stimulated mouse B lymphocytes rapidly increase their metabolic activity, including both oxidative phosphorylation and glycolysis compared to the non-activated state. However, in the absence of a costimulatory second signal that is provided by helper T cells, BCR-stimulated B cells progressively loose mitochondrial function and glycolytic capacity, leading to cell death [84]. Interestingly, anti-IgM-stimulated *bmb* B cells showed elevated OXPHOS and survival even after 3 days activation compared to similarly stimulated wt B cells. Further analyses are needed to resolve the detailed molecular mechanisms for this.

Over the years, we have studied the B cell response in IκBNS-deficient *bumble* mice with the objective to better understand how it regulates the NF-κB pathway and the differentiation of B cells in response to different stimuli. IκBNS remains a relatively understudied component of the NF-κB pathway, but we find it important as its loss in B cells clearly abrogates the T cell-independent antibody response [20,23]. We speculate that mutations in human *nfkbid* may underpin some common variable immunodeficiency syndrome (CVID) cases that remain unmapped, and we therefore believe that it is important to characterize its role in B cell development, activation, and differentiation in a model system as CVID patients often have defective responses to encapsulated bacteria [85].

## 5. Conclusions

In conclusion, our results demonstrate that the NF-κB regulator IκBNS plays an essential role in BCR- and TLR4-induced B cell survival, cell growth, cell cycle progression, and proliferation. Furthermore, our data suggest an opposing regulatory role for IκBNS in BCR and TLR4 pathways in terms of cell survival, and a complex and dual role during LPS- and anti-IgM-mediated proliferation and cell cycle regulation. These results broaden our knowledge of the role of IκBNS in the T-independent B cell responses.

## Figures and Tables

**Figure 1 cells-12-01229-f001:**
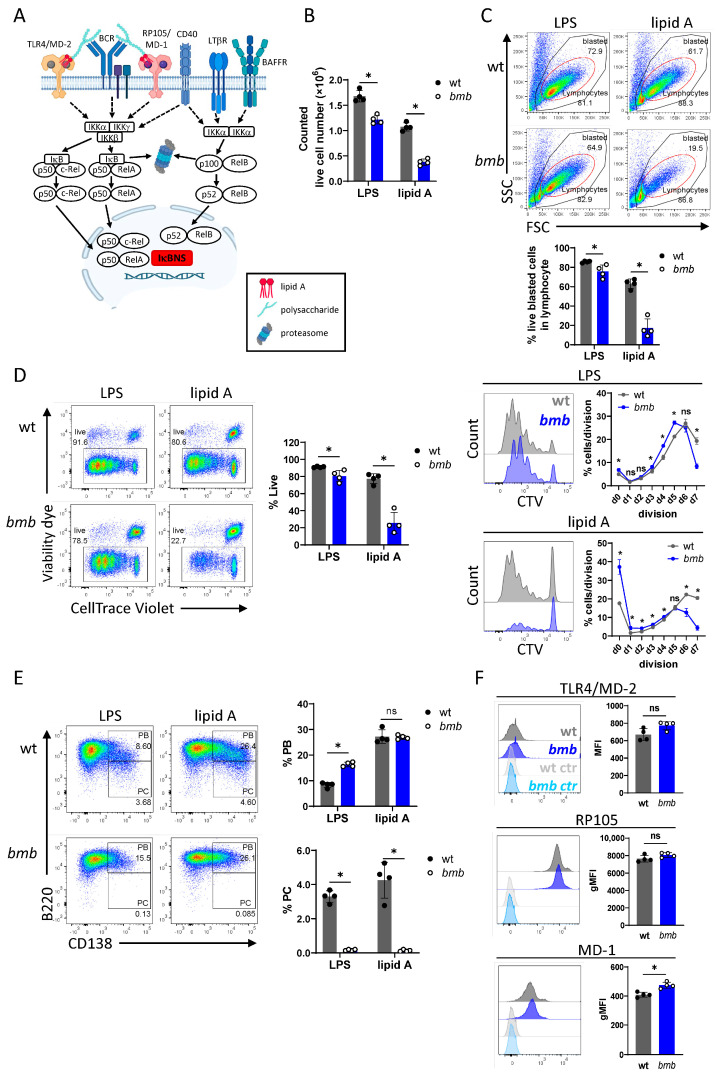
Impaired survival, proliferation, and differentiation of IκBNS-deficient B cells to TLR4 agonists, LPS, and lipid A. CellTrace Violet (CTV)-labeled wt and *bumble* (*bmb*) B cells were stimulated in vitro for three days with LPS or lipid A. (**A**) Schematic overview of classical and alternative NF-κB pathways with marking IκBNS in the nucleus. (**B**) Counted live cell numbers from LPS- or lipid A-stimulated wt (grey) and *bmb* (blue) cell cultures, using trypan blue exclusion assay. (**C**) Representative flow cytometry plots showing the gating strategy to identify lymphocytes based on their forward scatter (FSC) vs. side scatter (SSC) profile. Numbers adjacent to gates represent the frequency of gated cells in the parent (single cell) population. Red gates indicate blasting cells. Bar graphs showing the frequencies of live blasted cell populations within the lymphocyte gate. (**D**) Lymphocytes were further gated for live (viable dye^low^) proliferating cells. Bar graphs showing the frequency of viable cells. Representative overlayed histograms showing CTV dilution, corresponding graphs depicting the frequency of cells per each division from wt (grey) and *bmb* (blue) mice. (**E**) Representative FACS plots showing CD138 B220 co-staining for discriminating B220^+^CD138^+^ plasma blasts (PB) and B220^low/−^CD138^+^ plasma cells (PC). Bar graphs showing the percentages of PB and PC populations in wt (grey) and *bmb* (blue) cell cultures. (**F**) Histogram overlays and bar graphs representing the expression levels of TLR4/MD-2, CD180 (RP105), and MD-1 in freshly isolated wt (grey) and *bmb* (blue) B cells. FMO staining of wt (light grey) and *bmb* (light blue) B cells used as controls. Data were plotted as mean ± SD. Results are representative of four independent experiments with 3–4 mice for each genotype. Statistical significance was determined by Mann–Whitney *t*-test; ns (non-significant) for *p* > 0.05, * for *p ≤* 0.05. Figure 1A was created with BioRender.com.

**Figure 2 cells-12-01229-f002:**
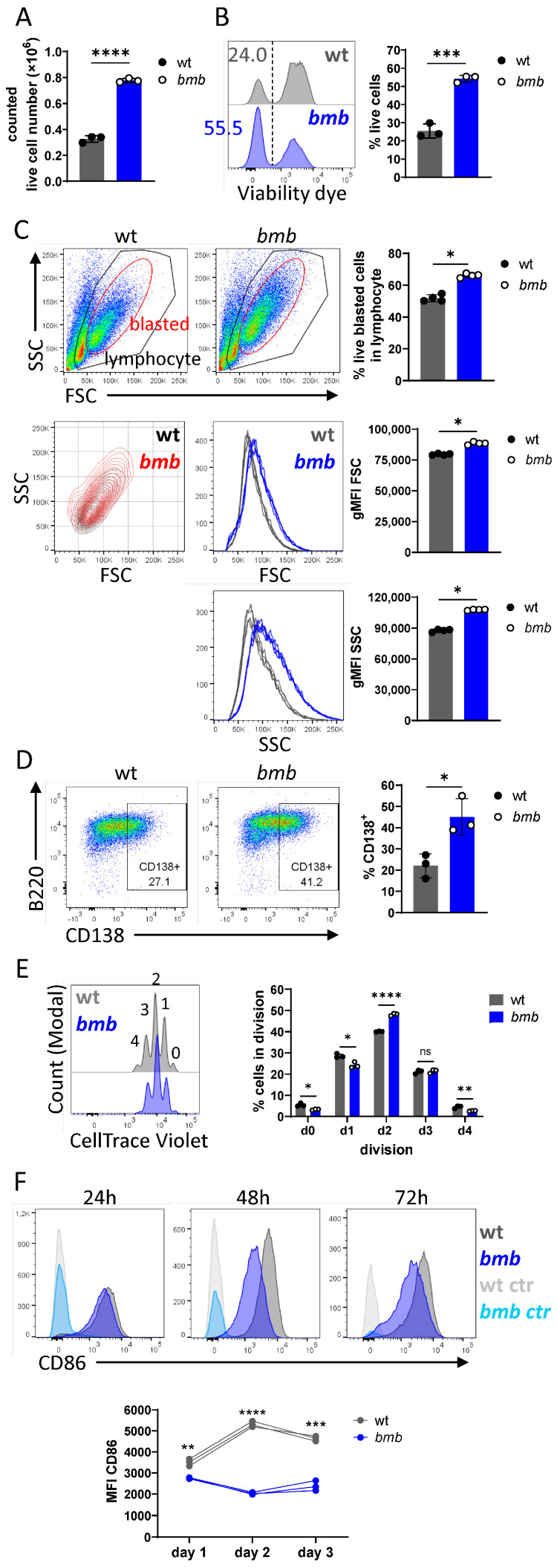
IκBNS-deficient B cells display improved survival, proliferation, and differentiation in response to in vitro BCR stimulation. Purified B cells from wt and *bmb* mice were stimulated in vitro with 10 μg/mL anti-IgM F(ab’)_2_ for 72 h. (**A**) Live cell numbers and (**B**) frequencies in wt (grey) or *bmb* (blue) cell cultures determined by trypan blue exclusion assay or flow cytometry, respectively. (**C**) Representative flow cytometry plots showing gating for lymphocyte (black gate), or blasted cells (red gate) in parent (live, single cell) population. Adjacent bar graphs representing the frequency of live blasted population in wt (grey) and *bmb* (blue) cultures. Overlayed contour plots showing blasted cell population in wt (black) and *bmb* (red), corresponding histogram overlays and bar graphs indicating cell size (FSC) and granularity (SSC) of blasted cells in wt (grey) and *bmb* (blue). (**D**) Representative FACS plots showing CD138 B220 co-staining, numbers adjacent to the gates indicating frequencies of CD138^+^ cells within gated live lymphocyte populations. Adjacent bar graphs showing frequencies of CD138^+^ populations in wt (grey) and *bmb* (blue) cell cultures. (**E**) CellTrace Violet (CTV)-labeled wt and *bmb* B cells were stimulated with anti-IgM in vitro for three days. Representative overlayed histograms showing CTV dilution of live cells, corresponding bar graphs depicting the frequencies of cells per each division from cell cultures. (**F**) wt and *bmb* B cells were stimulated in vitro with 10 μg/mL anti-IgM, or left unstimulated for 24, 48, and 72 h. Histogram overlays representing CD86 expression levels in stimulated wt (dark grey) and *bmb* (dark blue), or in non-stimulated wt (light grey) and *bmb* (light blue) B cells at each time point. Graph below showing mean fluorescence intensity (MFI) values for CD86 expression levels in stimulated wt (grey) and *bmb* (blue) B cells for each time point. All data were plotted as mean ± SD. Data are representative of four independent experiments with 3 or 4 mice for each strain. Statistical significance in A, B, D, E, and F was determined by unpaired *t*-test, and in C by Mann–Whitney *t*-test (ns (non-significant) for *p* > 0.05, * for *p ≤* 0.05, ** for *p ≤* 0.01, *** for *p ≤* 0.001, and **** for *p ≤* 0.0001).

**Figure 3 cells-12-01229-f003:**
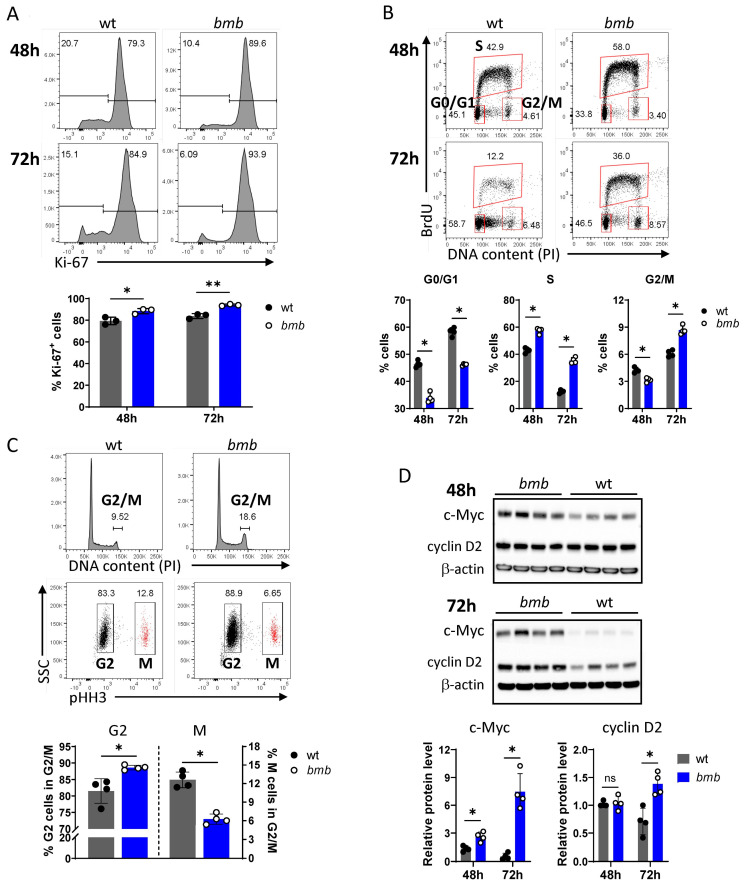
Accelerated cell cycle progression of BCR-stimulated IκBNS-deficient B cells is associated with a partial block in G2. Wt and *bumble* (*bmb*) B cells were stimulated in vitro with anti-IgM F(ab’)_2_. Cells were harvested after 48 or 72 h, and cell cycles were analyzed by flow cytometry. (**A**) Representative histograms of Ki-67 staining 48 or 72 h post stimuli in wt and *bmb* mice (upper panels). Bar graphs showing percentage of Ki-67^+^ B cells among live (viability dye^low^) cells (lower panel). (**B**) BCR-stimulated wt and *bmb* B cells were pulse labeled with BrdU for 1 h, then subsequently stained with anti-BrdU antibody and propidium iodide (PI). Representative flow cytometry plots showing the cell cycle profile of wt and *bmb* cells at 48 and 72 h post stimuli. G0/G1, S, and G2/M stages of the cell cycle are indicated by numbers adjacent to the gates, representing the frequency of cells in the corresponding cell cycle phases in the parent (live) population. Lower bar graphs representing the percentage of wt (grey bars) and *bmb* (blue bars) cells in each cell cycle phase at 48 and 72 h. (**C**) B cells from wt and *bmb* mice were cultured with anti-IgM F(ab’)_2_ for 72 h, and cell cycle analysis was performed by a combined staining of PI and pHH3. Frequency of G2/M cells in total live population is indicated (upper FACS panels). The fraction of G2 and M cells in G2/M was determined by pHH3 staining (lower FACS panels). Bar graphs showing the frequencies of cells in G2 and M within the G2/M population in wt (grey) and *bmb* (blue) mice. (**D**) Immunoblotting analysis of c-Myc, cyclin D2, and β-actin levels was performed using whole cell extracts of anti-IgM F(ab’)_2_-stimulated wt or *bmb* B cells, as indicated. Results are representative of two independent experiments with 3 or 4 mice. Data were plotted as mean ± SD, and statistical significance was determined by Mann–Whitney *t*-test; ns (non-significant) for *p* > 0.05, * for *p ≤* 0.05, ** for *p ≤* 0.01.

**Figure 4 cells-12-01229-f004:**
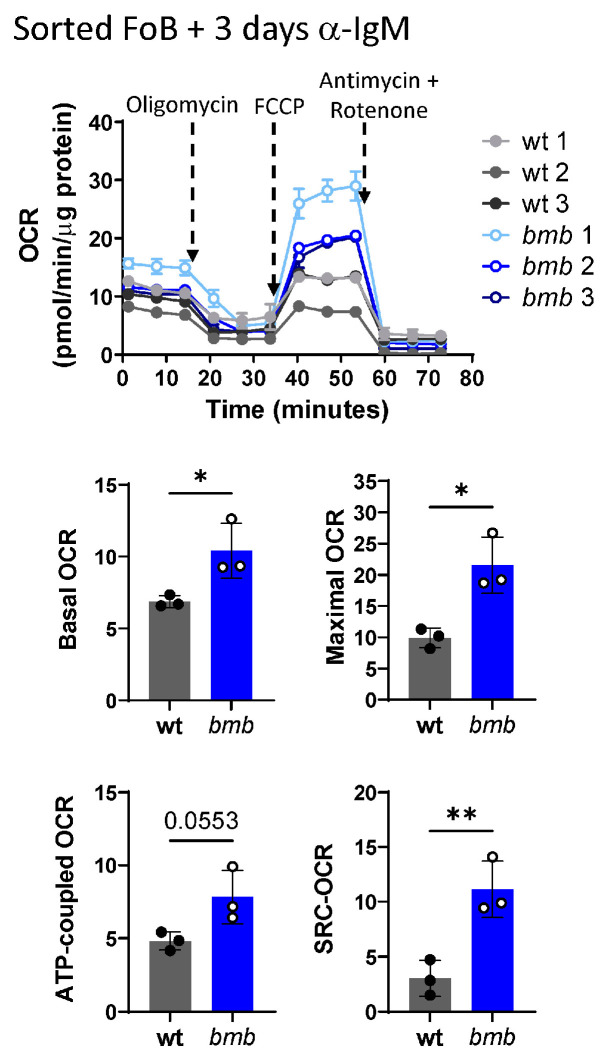
Enhanced oxidative phosphorylation in BCR-stimulated IκBNS-deficient follicular B cells. FACS-sorted follicular B cells (FoB) were stimulated with 10 μg/mL anti-IgM F(ab’)_2_ for 3 days. Oxygen consumption rates (OCR) were measured by Mito Stress Test under basal conditions and in response to the indicated mitochondrial inhibitors. Protein-normalized diverse OCR profiles of wt (gray) and *bmb* (blue) B cells are shown. Labeled arrows denoting injections of the inhibitors. Bar graphs showing calculated values for basal OCR, maximal OCR, Spare Respiratory Capacity, and ATP-coupled OCR in wt (grey) and *bmb* (blue). Statistical significance was determined by unpaired *t*-test (* *p* < 0.05, ** *p* < 0.01).

**Figure 5 cells-12-01229-f005:**
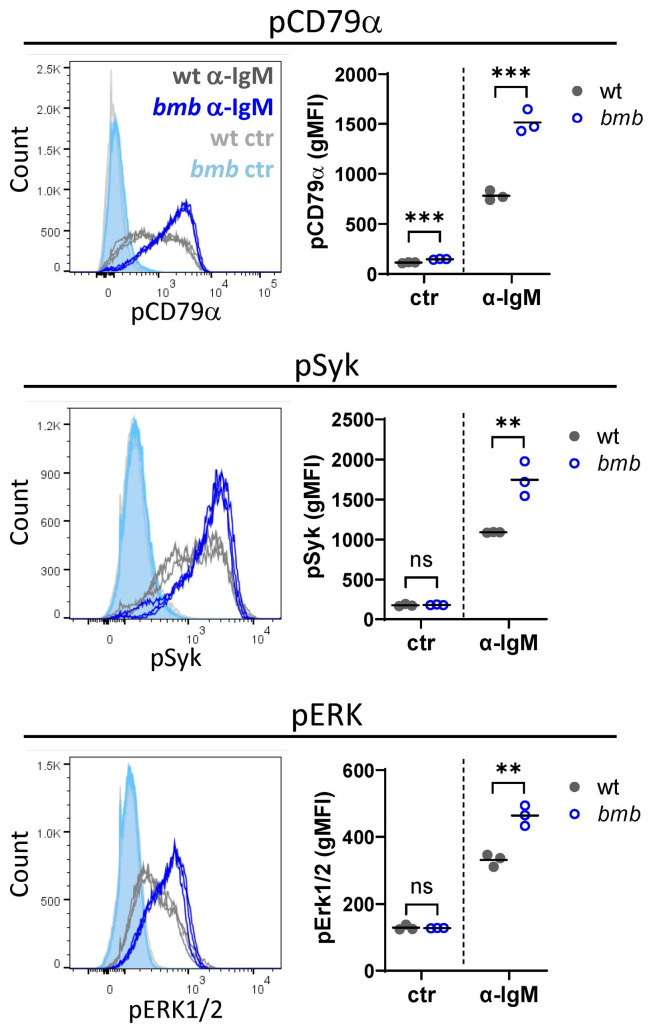
Enhanced BCR signaling in IκBNS-deficient *bumble* B cells. B cells from wt and *bumble* (*bmb*) mice were stimulated with 25 μg/mL anti-IgM F(ab’)_2_ for 5 min, and phosphoproteins were assessed by flow cytometry. Histogram overlays showing expression levels for pCD79α, pSyk, and pERK1/2 in non-stimulated wt (light grey), non-stimulated *bmb* (light blue), stimulated wt (dark grey), and stimulated *bmb* (dark blue) B cells. Adjacent graphs representing the corresponding geometric mean fluorescence intensity (*g*MFI) values. Graphs represent mean ± SD. Data are representative of two experiments with 3 or 4 mice of each strain. Statistical significance was determined by an unpaired *t*-test; ns (non-significant) for *p* > 0.05, ** for *p ≤* 0.01, *** for *p ≤* 0.001.

**Figure 6 cells-12-01229-f006:**
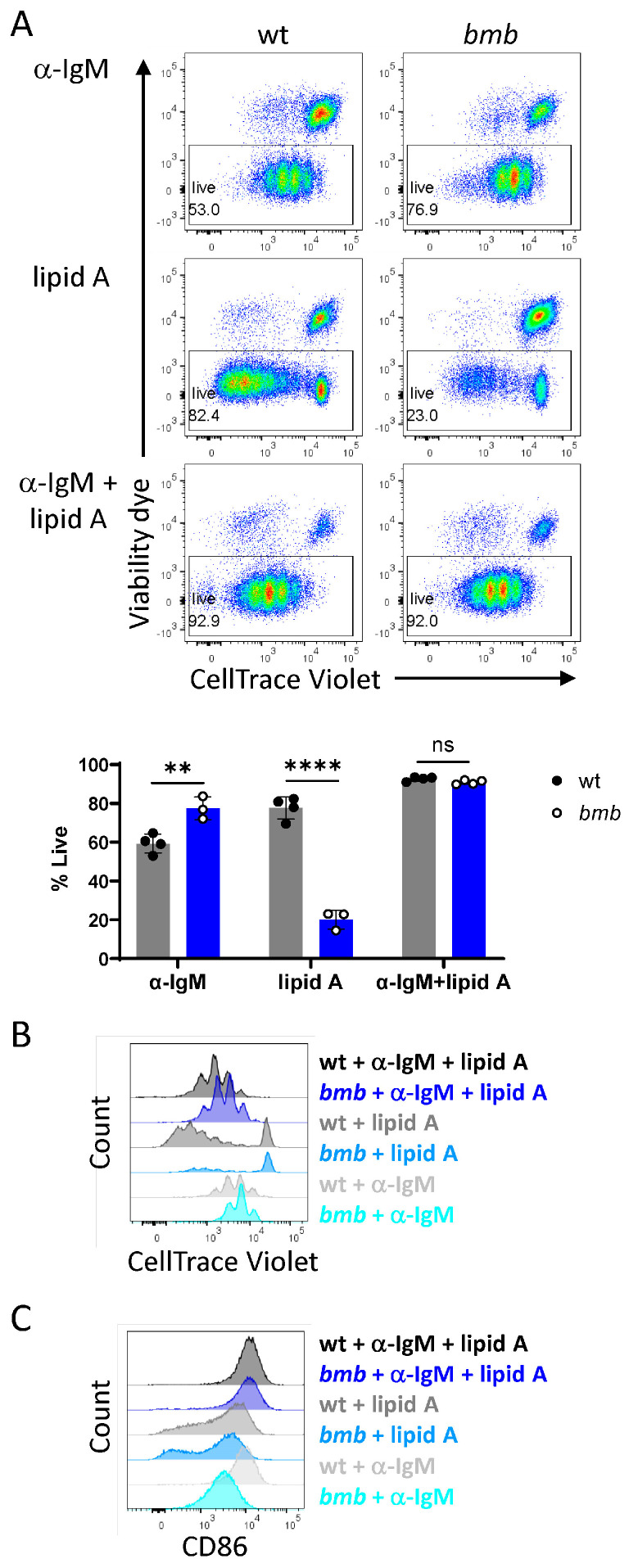
Combined BCR and TLR4 stimulation has a synergistic effect in IκBNS-deficient B cells regarding cell viability, proliferation, and activation. CTV-labeled wt and *bumble* (*bmb*) B cells were stimulated for 72 h with 10 μg/mL anti-IgM F(ab’)_2_, 1 μg/mL lipid A, or the combination of 10 μg/mL anti-IgM F(ab’)_2_ + 1 μg/mL lipid A, as indicated. (**A**) Representative flow cytometry plots showing CTV dilution versus live/dead staining. Numbers adjacent to the gates represent the frequency of gated population in parent (single cell) population. Bar graphs depict the frequency of live cells (viability dye^low^). Graphs represent mean ± SD; statistical significance was determined by unpaired *t*-test (ns (non-significant) for *p* > 0.05, ** for *p ≤* 0.01, **** for *p ≤* 0.0001). Overlayed histograms show the proliferation profile (**B**) or the expression levels of CD86 of stimulated cells (**C**) for wt and *bmb* mice for each condition, as indicated. Data are representative of two experiments with 4 or 6 mice of each strain.

**Figure 7 cells-12-01229-f007:**
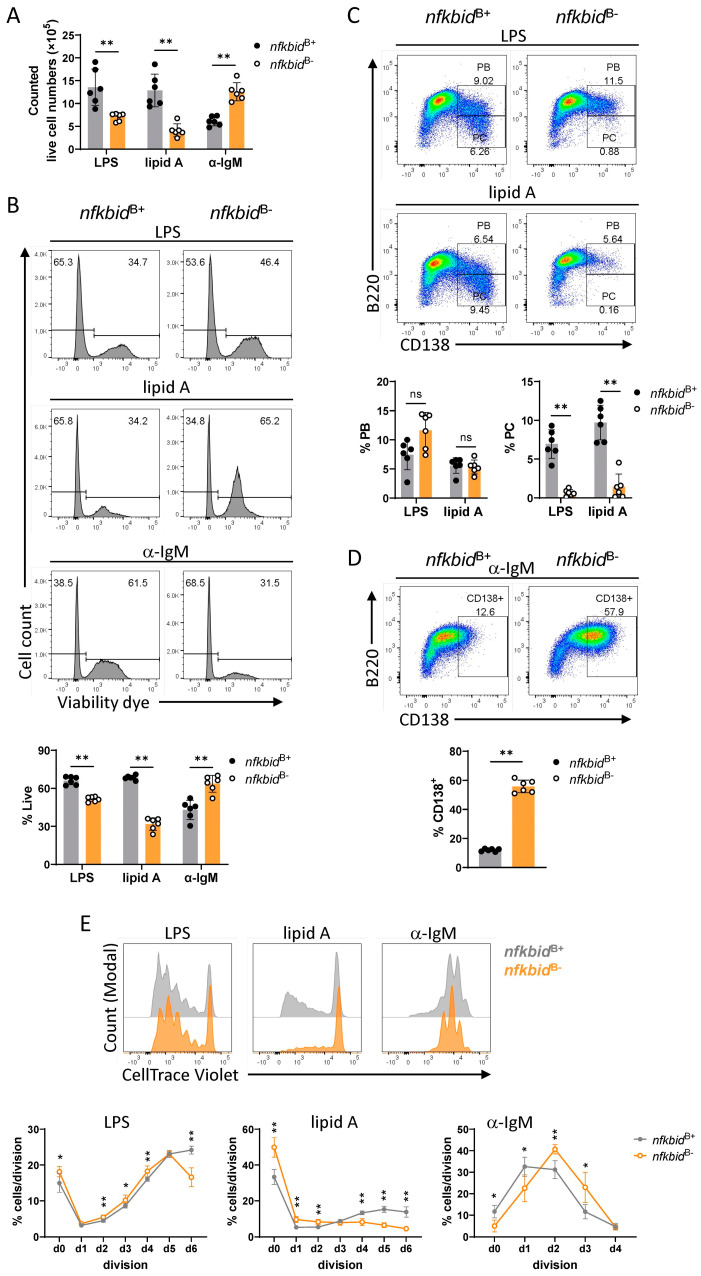
Reduced TLR4 response and enhanced BCR response in IκBNS conditional knock-out B cells. B cells from *nfkbid*^B+^ and *nfkbid*^B−^ mice were activated in vitro with LPS, lipid A, or anti-IgM F(ab’)_2_ for 72 h. After incubation, cells were harvested, and live cells were counted using trypan blue exclusion. (**A**) Bar graphs showing counted viable cell numbers in *nfkbid*^B+^ (grey) and *nfkbid*^B−^ (orange) cell cultures. (**B**) Representative flow cytometry histograms showing live/dead staining of *nfkbid*^B+^ and *nfkbid*^B−^ cells. Numbers above gates indicate percentages of live (viability dye^low^) and dead (viable dye^high^) cells in parent (singlets) population. Bar graphs showing frequencies of live cells in *nfkbid*^B+^ (grey) and *nfkbid*^B−^ (orange) cell cultures. (**C**) Representative flow cytometry plots showing staining for B220^+^CD138^+^ plasma blast (PB) and B220^low/−^CD138^+^ plasma cell (PC) populations in LPS- or lipid A-stimulated cell cultures at 72 h post stimuli. Bar graphs showing percentages of PB and PC populations. (**D**) Representative FACS plots showing B220 CD138 co-staining in anti-IgM F(ab’)_2_-stimulated *nfkbid*^B+^ (grey) and *nfkbid*^B−^ (orange) cell cultures at 72-h time point. Numbers adjacent to the gates represent frequencies of CD138^+^ cells within the parent (live lymphocyte) population. Bar graphs showing frequencies of CD138^+^ cells. (**E**) Representative overlayed histograms showing the proliferation profile of stimulated B cells from *nfkbid*^B+^ (grey) and *nfkbid*^B−^ (orange) cells at 72-h time point. Corresponding graphs depicting the frequencies of cells per each division for each stimulation. Graphs represent mean ± SD; statistical significance was determined by Mann–Whitney *t*-test (ns (non-significant) for *p* > 0.05, * *p* < 0.05, ** *p ≤* 0.01). Results are representative of two independent experiments with 4 or 6 mice of each genotype.

## Data Availability

All data needed to evaluate the conclusions in the paper are presented in the paper and the Appendix A. The data presented in this study are available on request from the corresponding author.

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
