# Peer review of "Enhanced B Cell Receptor Signaling Partially Compensates for Impaired Toll-like Receptor 4 Responses in LPS-Stimulated IκBNS-Deficient B Cells"

_cells, 2023, doi:10.3390/cells12091229_

Round 1

Reviewer 1 Report

The authors report that although IκBNS-deficient mice show impaired TLR4-mediated responses to LPS stimulation, the impairment is partially compensated by stimulation of B cell receptors.

The experiments were all carefully performed and their presentation will be sufficient. The text is also well written.

However, there are no in vivo experiments presented, and it is unclear how the results of this research will affect actual living organisms (even animal models).

Perhaps the following should be addressed in the DISCUSSION.

Even if the response of B cells after TLR4 stimulation is T cell-independent, for example, in the case of IκBNS-deficient mice, there may be an effect on T cells. Furthermore, what about the conditional deficient mice for B cells and what are the differences? What are your future plans in this direction?

What about the results of accelerated cell cycle after BCR stimulation, and how activation-induced cell death occurs when observed over a longer period of time?

What about the enhancement of oxidative phosphorylation following BCR stimulation shown in Figure 4, and what about mitochondrial function?

Reviewer 2 Report

Introduction

Line 38: Is this a mouse-specific mitogenic effect of LPS?

Figure 1: If possible, please make lipid A more clearly visible in the figure.

Methods

Please support the methodology part with more references

Line 126-127: the word flat is in duplicate.

Line 130: Was it not possible to use LPS and lipid A from the same E. coli strain for B cell stimulation

Line 136: please add the version of FlowJo.

Line 179: could the FC blocking with anti-CD16/CD32 have an impact on the analyzed activities of B cells?

219-221: please indicate whether the HRP-antibodies were monoclonal or polyclonal.

Results

Line 229: "PS induces proliferation" do you mean LPS ?

Line 249: If possible, an extra dot plot of non-stimulated cells may be included in Figure 1 to show the level of blastogenesis in non-stimulated lymphocytes

Figure 2 caption: line 325 please correct form into from

Discussion

What is the clinical relevance of the obtained results for the B cell response against capsular bacteria?
